# Interrogating the structure and function of the human voltage-gated proton channel (hH$_v$1) with a fluorescent noncanonical amino acid

**Emerson M Carmona, William N Zagotta\*, Sharona E Gordon\***

Department of Neurobiology and Biophysics, University of Washington, Seattle, United States

## eLife Assessment

This **important** study presents a **convincing** methodological approach to probe the structural features of the full-length human H$_v$1 channel as a purified protein. The method is supported by rigorous biochemical assays and spectral FRET analysis, which will interest biophysicists and physiologists studying H$_v$1 and other ion channels and membrane proteins. Overall, the work introduces an interesting labeling strategy and provides a methodology that is of value in investigating hH$_v$1 in particular and can be extended to other ion channels. The authors also provide preliminary observations regarding conformational changes induced by zinc.

**\*For correspondence:**
zagotta@uw.edu (WNZ);
seg@uw.edu (SEG)

**Competing interest:** The authors declare that no competing interests exist.

**Abstract** The human voltage-gated proton channel (hH$_v$1) is a dimer of voltage-sensor domains (VSDs) containing highly selective proton permeation pathways in each monomer. In addition to voltage, hH$_v$1 is regulated by other stimuli, including pH gradients, mechanical forces, and ligands, such as Zn$^{2+}$. Aside from the VSDs, this membrane protein contains an N-terminal domain and a C-terminal coiled-coil domain (CC) formed between the monomers. To address the need for direct measurements of conformational rearrangements in hH$_v$1, we developed a Förster resonance energy transfer (FRET) approach to measuring the conformational rearrangements in full-length hH$_v$1 purified from *E. coli*. We used genetic code expansion (GCE) to generate a library of 14 full-length hH$_v$1 constructs, each incorporating the fluorescent noncanonical amino acid acridon-2-ylalanine (Acd) at a different site throughout the various structural domains. Following the expression and purification of these hH$_v$1-Acd proteins, we found that 12 sites yielded stable and functional proton-permeable channels. The fluorescence properties of Acd at each site showed small site-specific differences. Furthermore, we measured site-specific FRET efficiencies from tryptophan (Trp) and tyrosine (Tyr) to Acd in the hH$_v$1-Acd proteins and found results consistent with correct folding in detergent micelles. Finally, the addition of Zn$^{2+}$ produced reversible changes in FRET, with affected residues clustered on the intracellular side of the channel.

## Introduction

The human voltage-gated proton channel (hH$_v$1) is a membrane protein that forms a proton selective channel (*Sasaki et al., 2006*; *Ramsey et al., 2006*). Despite its physiological importance, its molecular mechanisms are still poorly understood, including the precise structural identity of the permeation pathway and the gating mechanism of voltage (*Cherny et al., 1995*; *Carmona et al., 2018*), pH gradients (*Cherny et al., 1995*; *Carmona et al., 2021*), membrane stretch (*Pathak et al.,*

*2016*), and ligands, such as the classical inhibitor $Zn^{2+}$ (*Cherny and DeCoursey, 1999*; *DeCoursey and Cherny, 1994*). Understanding these processes requires connecting the functional properties to the $hH_v1$ structure.

Each monomer of $hH_v1$ dimers includes an intracellular N-terminal domain of unknown function, a transmembrane voltage-sensor domain (VSD), and an intracellular coiled-coil (CC), which forms the primary intersubunit interface (*Lee et al., 2008*; *Koch et al., 2008*). Current structural models, however, have been obtained from truncated or chimeric channel constructs (*Takeshita et al., 2014*; *Bayrhuber et al., 2019*), and thus, it is not clear whether they correspond to physiological or functional conformations. Moreover, the multiple kinetic components of $H_v1$ gating currents suggest that the protein's resting state comprises an ensemble of conformations (*Carmona et al., 2018*; *Carmona et al., 2021*), which is consistent with the conformational flexibility inferred from structural approaches (*Bayrhuber et al., 2019*; *Li et al., 2015*; *Han et al., 2022b*). Because protein function arises from this distribution of conformational states (*Henzler-Wildman and Kern, 2007*), approaches to directly measure structural heterogeneity in the full-length $hH_v1$ are needed.

An attractive approach to interrogate the structural heterogeneity of $hH_v1$ is fluorescence spectroscopy, which offers high sensitivity at nanomolar protein concentrations and compatibility with near-physiological conditions. Fluorescence can report on local environment and accessibility changes, as well as quantify distance changes through Förster resonance energy transfer (FRET) (*Lakowicz, 2006*). Importantly, time-resolved FRET measured using lifetimes preserves information about conformational heterogeneity on the nanosecond timescale, which is lost in steady-state measurements due to the averaging process (*Gordon et al., 2018*). Together, these capabilities make fluorescence, especially time-resolved FRET, a powerful method to connect $hH_v1$ structure and function (*Zagotta et al., 2024*).

A few obstacles remain to directly measuring conformational heterogeneity of $hH_v1$. First, a method to express and purify stable and functional protein is needed, which is challenging for membrane proteins. Second, site-specific fluorophore labeling is difficult at buried sites, resulting in low labeling efficiency. Third, traditional fluorophores are bulky and have long linkers (*Han et al., 2022b*), resulting in protein structure perturbation and heterogeneity dominated by the properties of the label (*Taraska et al., 2009*). A recent method was optimized for robust expression and purification of full-length, functional $hH_v1$ in *E. coli* (*Carmona et al., 2025a*; *Carmona et al., 2025b*). The remaining obstacles can be overcome by labeling the protein with the fluorescent noncanonical amino acid acridon-2-ylalanine (Acd) using genetic code expansion (GCE) (*Speight et al., 2013*; *Sungwienwong et al., 2017*; *Jones et al., 2021*). This technology allows site-specific labeling during translation by reassigning an amber stop codon using an orthogonal aminoacyl-tRNA synthetase (RS)/tRNA (*Liu and Schultz, 2010*; *Young et al., 2010*; *Figure 1A*). Acd's small size and minimal linker reduce perturbations and heterogeneity, and its favorable photophysics support both steady-state and time-resolved measurements (*Zagotta et al., 2021*). Here, we incorporated Acd across all $hH_v1$ structural domains and purified 12 of 14 constructs as stable, functional proteins. We then used Acd's environmental sensitivity and FRET from tryptophan (Trp)/tyrosine (Tyr) to Acd to validate the protein folding and report $Zn^{2+}$-dependent conformational changes. These data establish a platform for interrogating conformational heterogeneity and dynamics in the full-length $hH_v1$.

## Results

### We generated a library of $hH_v1$ constructs with Acd incorporated across the different structural domains

Because available $H_v1$ structural models were produced from either truncated or modified proteins (*Takeshita et al., 2014*; *Bayrhuber et al., 2019*), we used an AlphaFold model of the full-length, dimeric $hH_v1$ to guide Acd site selection (*Jumper et al., 2021*; *Kim et al., 2025*; *Figure 1B* and *Source data 1*). The model recapitulates the expected $hH_v1$ architecture with high confidence: S0 as a short helix parallel to the membrane, S1-S4 transmembrane segments forming a canonical VSD, and a C-terminal helix extending from S4 that forms the CC intersubunit interface. In addition, two helices in the N-terminal domain (P1 and P2) were predicted (*Figure 1C*), albeit with low confidence. The dimer interface in the membrane is primarily formed by S4-S4 contacts (*Figure 1—figure supplement 1A*), although S1-S1 contacts have been observed experimentally (*Lee et al., 2008*; *Li et al., 2015*). This model appears to represent an intermediate state, with the first S4 positive charged residue R208

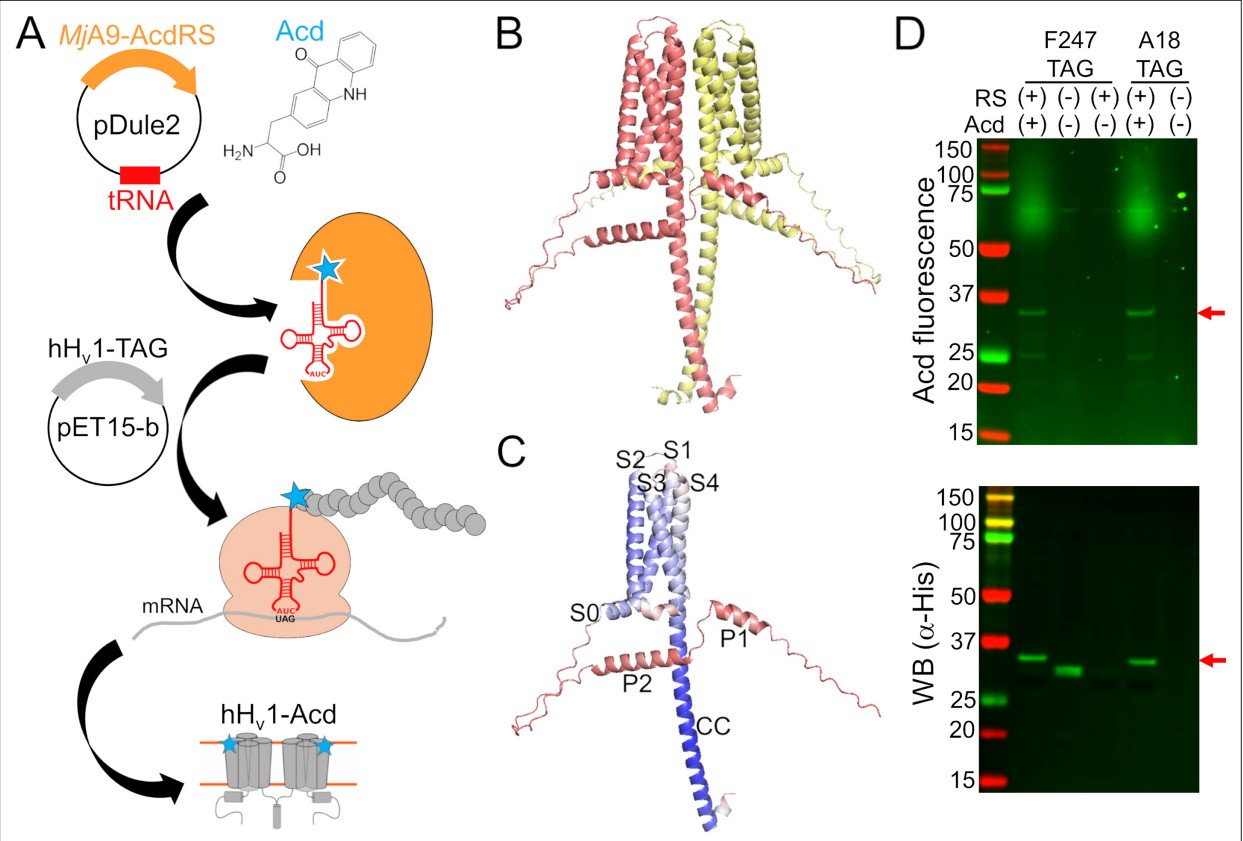

**Figure 1.** Human voltage-gated proton channel (hHᵥ1) was labeled with the fluorescent noncanonical amino acid acridon-2-ylalanine (Acd) by genetic code expansion. (**A**) Cartoon showing the components of the genetic code expansion system used to incorporate Acd via amber codon suppression in hHᵥ1. (**B**) AlphaFold structural model of the dimer hHᵥ1 colored by subunit. (**C**) AlphaFold structural model of a single hHᵥ1 subunit colored by the predicted Local Distance Difference Test (pLDDT). Model confidence was represented by a gradient from red to blue, indicating low to high pLDDT. (**D**) In-gel fluorescence and Western blot to evaluate Acd incorporation. An Acd fluorescence band corresponding to the molecular weight of full-length hHᵥ1 (red arrow) was observed in cellular extracts after separation by SDS-PAGE only when cells contained the *Mj*A9-AcdRS/tRNA plasmid (RS) and were grown in the presence of 1 mM Acd (top). The Western blot against the His-tag present at the N-terminus of hHᵥ1 confirmed that the band corresponded to the recombinant protein (bottom). Note the hHᵥ1 truncated product produced when the amber stop codon is replacing F247 in the absence of the aminoacyl-tRNA synthetase/tRNA plasmid and Acd.

The online version of this article includes the following source data and figure supplement(s) for figure 1:

**Source data 1.** Original files for acridon-2-ylalanine (Acd) fluorescence and Western blot showed in *Figure 1* indicating the relevant bands.

**Source data 2.** Original files for acridon-2-ylalanine (Acd) fluorescence and Western blot showed in *Figure 1*.

**Figure supplement 1.** AlphaFold dimer human voltage-gated proton channel (hHᵥ1) structural model.

proximal to D112 in the selectivity filter, and the third S4 positive charged residue R211 proximal to F150 in the charge-transfer center (***Figure 1—figure supplement 1B***). Although the model's accuracy and functional state are unknown, we used it as a rough starting point for selecting positions to incorporate Acd.

To test the feasibility and specificity of Acd incorporation in hHᵥ1, we initially selected two positions in the intracellular domains: A18 in the N-terminal domain and F247 in the CC. For each position, the wild-type codon was replaced with an amber stop codon (TAG). Bacteria co-transformed with the Acd RS/tRNA pair and hHᵥ1-TAG constructs expressed fluorescent full-length protein only when Acd was present in the culture medium (***Figure 1D***). We next generated a library of 14 constructs in which Acd was incorporated into every structural domain: the N-terminal domain, each helix of the VSD, and the CC (***Figure 2A–B***). Remarkably, most of the 14 hHᵥ1-Acd proteins expressed at levels comparable to the control (No TAG) (***Figure 2C***, ***Figure 2—figure supplement 1***). As expected, several hHᵥ1-Acd proteins also produced TAG-truncated products (***Figure 2C***). Increasing the Acd concentration in the *E. coli* growth medium increased the protein expression but did not reduce the fraction of truncated

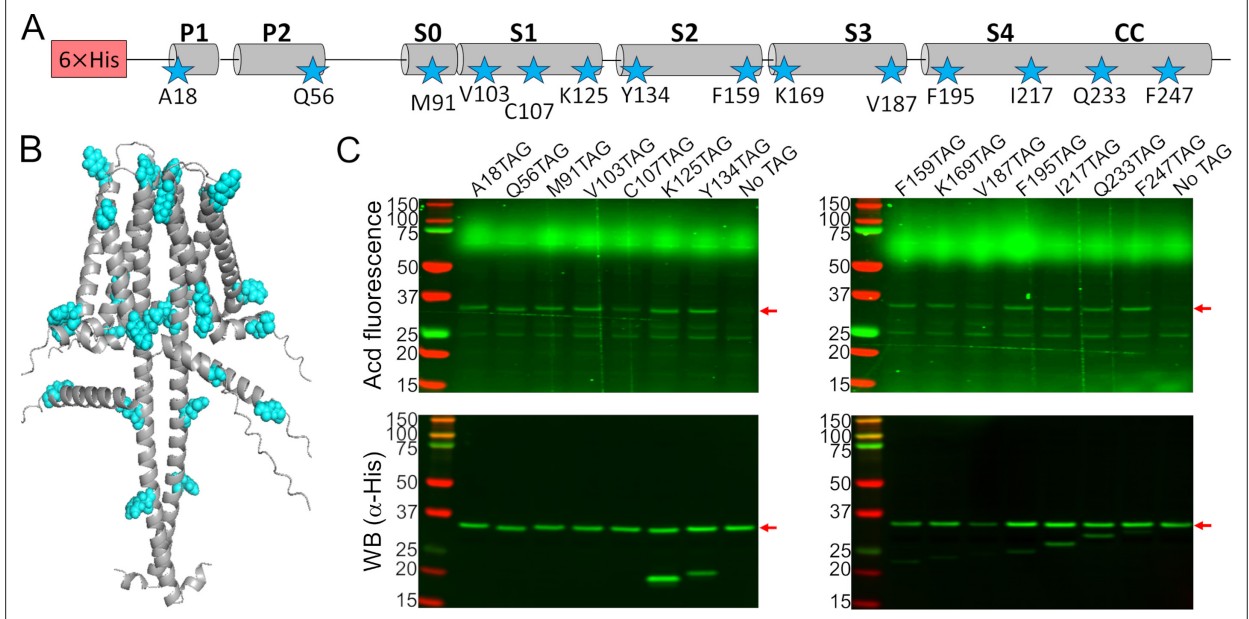

**Figure 2.** Acridon-2-ylalanine (Acd) was incorporated into 14 positions in the human voltage-gated proton channel (hH$_v$1) sequence. (**A**) Cartoon showing the amino acids selected to be replaced by an amber stop codon in the hH$_v$1 secondary structure to incorporate Acd (stars). (**B**) AlphaFold dimer hH$_v$1 structural model with the amino acids selected to be replaced by Acd as cyan spheres. (**C**) Acd fluorescence gel (top) and Western blot (bottom) of cellular extracts after separation by SDS-PAGE showing the expression of the Acd-labeled hH$_v$1 mutants (red arrows). The shorter detected bands in the WB for some constructs are produced by translation termination. The No TAG lane contains an extract from cells grown under identical conditions (in the presence of Acd and the aminoacyl-tRNA synthetase/tRNA pair) and expressing hH$_v$1 without an amber stop codon.

The online version of this article includes the following source data and figure supplement(s) for figure 2:

**Source data 1.** Original files for acridon-2-ylalanine (Acd) fluorescence and Western blot showed in *Figure 2* and analyzed in *Figure 2—figure supplement 1* indicating the relevant bands.

**Source data 2.** Original files for acridon-2-ylalanine (Acd) fluorescence and Western blot showed in *Figure 2* and analyzed in *Figure 2—figure supplement 1*.

**Figure supplement 1.** The protein yield of human voltage-gated proton channel (hH$_v$1)-acridon-2-ylalanine (Acd) depended on the position of Acd incorporation.

**Figure supplement 1—source data 1.** OD$_{600}$ measured at the end of the cultures expressing human voltage-gated proton channel (hH$_v$1)-acridon-2-ylalanine (Acd) analyzed in *Figure 2—figure supplement 1*.

**Figure supplement 2.** The protein yield of human voltage-gated proton channel (hH$_v$1)-A18Acd was proportional to the concentration of acridon-2-ylalanine (Acd) in the culture medium.

**Figure supplement 2—source data 1.** Original files for acridon-2-ylalanine (Acd) fluorescence and Western blot showed and analyzed in *Figure 2—figure supplement 2* indicating the relevant bands.

**Figure supplement 2—source data 2.** Original files for acridon-2-ylalanine (Acd) fluorescence and Western blot showed and analyzed in *Figure 2—figure supplement 2*.

**Figure supplement 2—source data 3.** OD$_{600}$ measured at the end of the cultures expressing hH$_v$1-A18Acd analyzed in *Figure 2—figure supplement 2*.

**Figure supplement 3.** The truncation of human voltage-gated proton channel (hH$_v$1)-K125Acd was not decreased at higher concentrations of acridon-2-ylalanine (Acd) in the culture medium.

**Figure supplement 3—source data 1.** Original files for acridon-2-ylalanine (Acd) fluorescence and Western blot showed and analyzed in *Figure 2—figure supplement 3* indicating the relevant bands.

**Figure supplement 3—source data 2.** Original files for acridon-2-ylalanine (Acd) fluorescence and Western blot showed and analyzed in *Figure 2—figure supplement 3*.

**Figure supplement 3—source data 3.** OD$_{600}$ measured at the end of the cultures expressing hH$_v$1-K125Acd analyzed in *Figure 2—figure supplement 3*.

protein (*Figure 2—figure supplements 2 and 3*). One of the constructs, hH$_v$1-C107Acd, showed a weak in-gel fluorescence signal, although the Western blot signal was comparable to that of other constructs.

## We successfully purified stable and functional hH$_v$1 labeled with Acd at 12 of the 14 selected positions

We attempted to purify all expressed hH$_v$1-Acd constructs by immobilized metal affinity chromatography using the detergent Anzergent 3–12 (*Carmona et al., 2025a*; *Carmona et al., 2025b*). For 12 of 14 constructs, purification yielded a single fluorescent band on SDS-PAGE (*Figure 3A*, *Figure 3—figure supplement 1*). In contrast, no purified protein was detected for the V187Acd or Q233Acd constructs (*Figure 3—figure supplement 1*). Because hH$_v$1-Q233Acd was robustly expressed (*Figure 2C*), its loss during purification suggests reduced stability. The hH$_v$1-Acd yields were construct-dependent (*Figure 3B*), and the TAG-truncated products did not co-purify with the full-length protein (*Figure 3A*, *Figure 3—figure supplement 1*).

We then evaluated the stability and homogeneity of the hH$_v$1-Acd proteins using fluorescence-detection size exclusion chromatography (FSEC) (*Kawate and Gouaux, 2006*). Similar to the control (*Figure 3—figure supplement 2*), most hH$_v$1-Acd proteins showed a predominant main peak (~11.5 mL), which corresponds to the dimer (*Figure 3—figure supplement 2*), followed by a minor peak at higher elution volumes (*Figure 3C* and *Table 1*). The proportion of these two peaks varied between constructs (*Figure 3—figure supplement 3*). To determine whether the purified hH$_v$1-Acd proteins were functional, we reconstituted the FSEC fractions containing both peaks in asolectin liposomes and assessed their function using a liposome proton flux assay (*Carmona et al., 2025a*; *Lee et al., 2009*). All 12 reconstituted constructs produced ACMA fluorescence quenching upon valinomycin addition, indicating that our purified proteins formed functional proton-permeable channels (*Figure 3D*, *Figure 3—figure supplement 4*). These results highlight the versatility of GCE for incorporating Acd as a fluorescent probe for structural studies of hH$_v$1.

## Acd was relatively insensitive to the hydrophobicity of its local environment

To explore how the local environment can alter Acd's spectral properties, we measured the emission spectra of the free amino acid in different solvents (*Figure 4A* and *Table 2*). The environmental sensitivity of Acd has been studied previously (*Jones et al., 2021*; *Zagotta et al., 2021*; *Szymańska et al., 2003*), but we expanded this work here by using the aprotic solvent ethyl acetate (EtAc) and a series of alcohols with varying alkyl chain lengths. Consistent with a general solvent effect produced by polarity (*Lakowicz, 2006*), the emission spectrum was blue-shifted by approximately 20 nm in EtAc, the least polar solvent measured, relative to water (*Figure 4A* and *Table 2*). Alcohols produced a consistently smaller blue shift, independent of their alkyl-chain length (*Figure 4A*). The spectrum in Buffer-H2, our standard experimental solution containing detergent micelles, was only very slightly blue-shifted relative to water (*Figure 4A*), suggesting that Acd did not substantially partition into the hydrophobic core of the detergent micelles. We also tested whether solvent polarity affects the fluorescence lifetime of Acd using time-correlated single photon counting (TCSPC). The fluorescence lifetime of Acd showed a similar trend to the spectral shifts (*Figure 4B*): EtAc shortened the lifetime markedly, Acd exhibited shorter lifetimes in alcohols than in water, and Acd in Buffer-H2 had a lifetime similar to that measured in water.

The spectral blue-shift and short lifetime of Acd in EtAc suggest that Acd may be able to report on the local environment at different sites in hH$_v$1, especially those that are surface-exposed versus those that face the lipid core. The emission spectra of the hH$_v$1-Acd proteins showed small, site-specific shifts (*Figure 4C* and *Table 3*). The local environment of Acd in hH$_v$1 spanned the range between those observed for free Acd in Buffer-H2 and in alcohols, with Q56Acd in the N-terminal intracellular domain and C107Acd in S1 showing the most red-shifted and blue-shifted spectra, respectively (*Figure 4C*). Consistent with the spectra, the fluorescence lifetimes of the hH$_v$1-Acd proteins fell between those measured for free Acd in Buffer-H2 and in alcohols, with Q56Acd exhibiting the slowest, and C107Acd the fastest lifetimes (*Figure 4D*). While small, the differences in spectral properties of Acd in hH$_v$1 agree with their predicted location in the structural model, with C107 being the site closest to the detergent micelles' hydrophobic core. The low sensitivity to polarity and pH changes (*Schulman and*

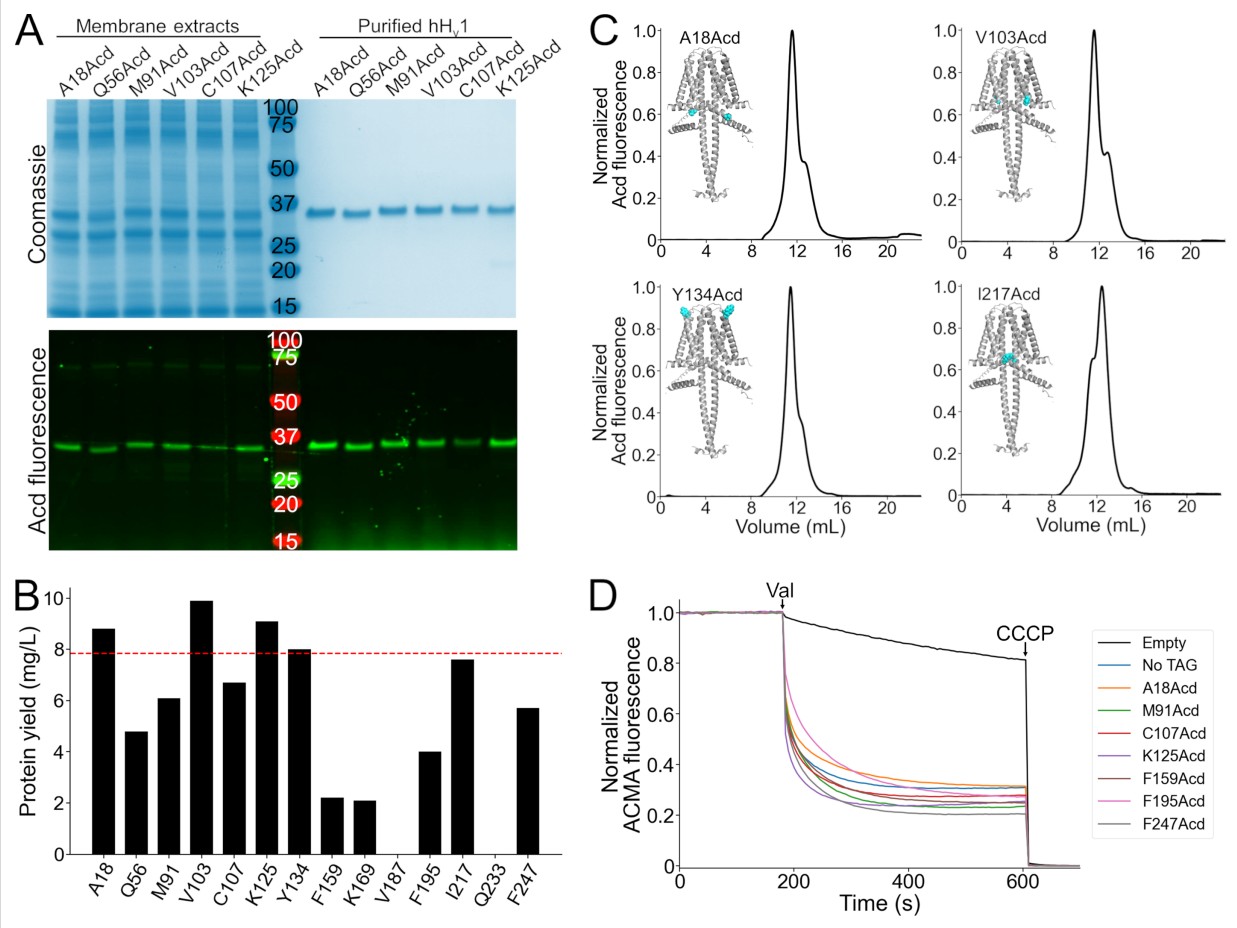

**Figure 3.** Purification and functional measurements of human voltage-gated proton channel (hH$_v$1)-acridon-2-ylalanine (Acd) proteins. (**A**) Coomassie-stained (top) and Acd fluorescence (bottom) gels after separation by SDS-PAGE showing representative samples of the membrane extracts after solubilization (left) and the final purified protein after immobilized metal affinity chromatography (right). (**B**) Protein yield of the hH$_v$1 protein with Acd replacing the amino acid at the indicated position. Note that V187Acd and Q233Acd contained no detectable protein after purification. The red line indicates the protein yield obtained from hH$_v$1 without an amber stop codon in the presence of Acd and the aminoacyl-tRNA synthetase/tRNA pair (No TAG). (**C**) Representative fluorescence-detection size-exclusion chromatograms of the purified hH$_v$1 proteins with Acd incorporated at the indicated amino acid position. Insets: AlphaFold dimer models, with the amino acid replaced by Acd shown as cyan spheres. (**D**) Representative liposome proton flux assay of asolectin proteoliposomes containing the indicated hH$_v$1 protein. All purified proteins produced 9-Amino-6-chloro-2-methoxyacridine (ACMA) fluorescence quenching after the addition of valinomycin (Val). The protonophore carbonyl cyanide m-chlorophenyl hydrazone (CCCP) was added at the end of the experiment as a control. The No TAG sample corresponds to proteoliposomes containing hH$_v$1 without an amber stop codon expressed in the presence of Acd and the aminoacyl-tRNA synthetase/tRNA pair.

The online version of this article includes the following source data and figure supplement(s) for figure 3:

**Source data 1.** Original files for acridon-2-ylalanine (Acd) fluorescence and Western blot showed in *Figure 3* indicating the relevant bands.

**Source data 2.** Original files for acridon-2-ylalanine (Acd) fluorescence and Western blot showed in *Figure 3*.

**Source data 3.** Original data for chromatograms showed in *Figure 3*, *Figure 3—figure supplement 3*.

**Source data 4.** Original data for the liposome proton flux assays showed in *Figure 3*, *Figure 3—figure supplement 4*.

**Figure supplement 1.** SDS-PAGE of the human voltage-gated proton channel (hH$_v$1) protein samples after purification.

**Figure supplement 1—source data 1.** Original files for acridon-2-ylalanine (Acd) fluorescence and Western blot showed in *Figure 3—figure supplement 1* indicating the relevant bands.

**Figure supplement 1—source data 2.** Original files for acridon-2-ylalanine (Acd) fluorescence and Western blot showed in *Figure 3—figure supplement 1*.

**Figure supplement 2.** The main human voltage-gated proton channel (hH$_v$1) peak in the size exclusion chromatogram corresponds to the dimer.

**Figure supplement 2—source data 1.** Original data for chromatograms showed in *Figure 3—figure supplement 2*.

*Figure 3 continued on next page*

*Figure 3 continued*

**Figure supplement 3.** Fluorescence-detection size exclusion chromatography of human voltage-gated proton channel (hH$_v$1)-acridon-2-ylalanine (Acd) proteins.

**Figure supplement 4.** The purified human voltage-gated proton channel (hH$_v$1)-acridon-2-ylalanine (Acd) proteins were functional proton channels.

*Sturgeon, 1977*) of Acd in different hH$_v$1 local environments makes it well suited for measuring FRET (*Zagotta et al., 2021*; *Gordon et al., 2024*).

## FRET between Trp/Tyr and Acd in hH$_v$1 suggested that the proteins are properly folded

Trp and Tyr have fluorescence emission spectra that overlap with the Acd absorption spectrum and, therefore, can act as FRET donors to Acd, with relatively short $R_0$ values (23.5 Å for Trp; 20.9 Å for Tyr) (*Speight et al., 2013*; *Figure 5A*). Each hH$_v$1 monomer contains four Trp and four Tyr residues (yellow spheres in *Figure 5B*). In this structural model, the Trp residues are located in the N-terminal domain (W4, W38, and W45) and midway along transmembrane segment S4 in the VSD (W207). The Tyr residues are located in the N-terminal domain (Y35 and Y42) and near the extracellular end of transmembrane segment S2 (Y134 and Y141).

To quantify Trp/Tyr to Acd FRET, we used spectral FRET analysis (*Clegg, 1992*). The total emission spectrum from Trp/Tyr and Acd was collected (*Figure 5C*). The Acd emission spectrum was extracted by subtracting a scaled Trp/Tyr spectrum collected from control wild-type protein ($F_{280,NoTAG}$) (*Figure 5D*, *Figure 5—figure supplement 1*). For each Acd-containing construct, the ratio of the extracted spectrum ($F_{280} - F_{280,NoTAG}$) to the Acd spectrum with direct excitation ($F_{370}$) was calculated as *Ratio A* (*Figure 5D and E*). Because *Ratio A* is not wavelength-dependent, it reports the linearity of the detectors and the absence of significant contamination from other sources of fluorescence. The *Ratio A* component caused by the direct excitation of Acd (termed *Ratio A$_0$*) was measured with free Acd (*Zheng et al., 2003*; *Figure 5E*, *Figure 5—figure supplement 2*). The difference between *Ratio A* and *Ratio A$_0$* is directly proportional to the FRET efficiency.

As expected, the *Ratio A* values for our hH$_v$1-Acd proteins were largely flat across emission wavelengths for our hH$_v$1-Acd proteins (*Figure 5E*). For all Acd sites, *Ratio A* was greater than *Ratio A$_0$* (dashed line in *Figure 5E*), reflecting FRET between Trp/Tyr and Acd at every site in hH$_v$1. Most *Ratio A* values were distributed between 0.60 and 0.80, with two clear exceptions: A18Acd (blue) and F247Acd (magenta) (*Figure 5E*). A18Acd showed the highest *Ratio A*, consistent with the multiple

**Table 1.** Size-exclusion chromatography elution volumes of the human voltage-gated proton channel (hH$_v$1) proteins. Elution volumes were measured as the location of the maximum value of absorbance at 280 nm in the chromatograms.

| hH$_v$1 sample | Elution volume (mL) |
|---|---|
| No TAG | 11.5 |
| A18Acd | 11.6 |
| Q56Acd | 11.7 |
| M91Acd | 11.9 |
| V103Acd | 11.5 |
| C107Acd | 12.5 |
| K125Acd | 11.7 |
| Y134Acd | 11.5 |
| F159Acd | 11.5 |
| K169Acd | 11.5 |
| F195 Acd | 11.5 |
| I217Acd | 12.4 |
| F247Acd | 13.1 |

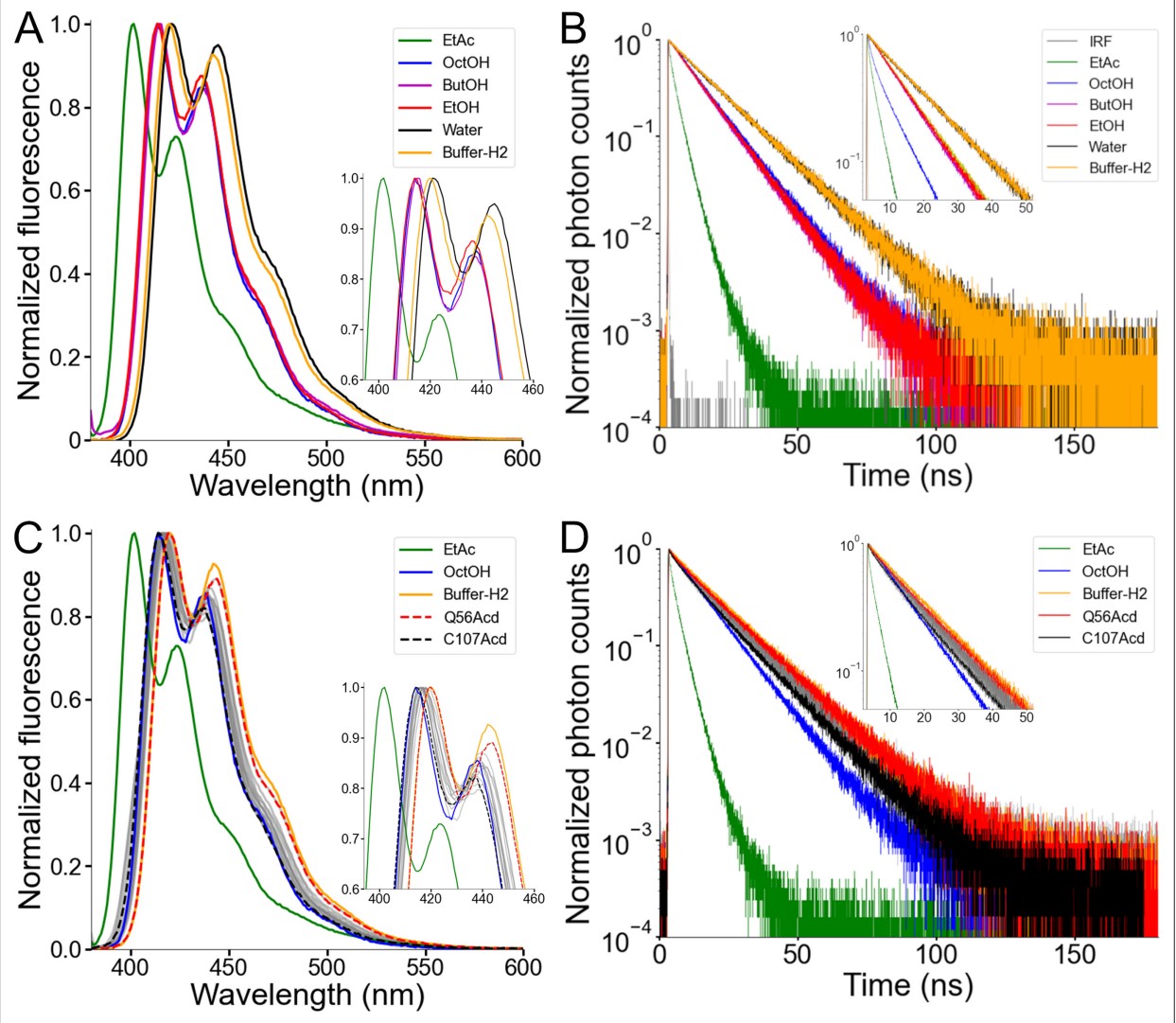

**Figure 4.** Acridon-2-ylalanine (Acd) showed a low environmental sensitivity in human voltage-gated proton channel (hH$_v$1). (**A**) Normalized fluorescence emission spectra of the Acd amino acid dissolved in the indicated solvent (Excitation = 370 nm). EtAc: ethyl acetate, OctOH: 1-octanol, ButOH: 1-butanol, EtOH: ethanol. (**B**) Normalized fluorescence decay of the Acd amino acid dissolved in the indicated solvent. IRF: Instrument Response Function. (**C**) Normalized fluorescence emission spectra of the hH$_v$1-Acd proteins in Buffer-H2 (gray). The Acd amino acid spectra in selected solvents are also shown for reference. Q56Acd and C107Acd spectra are shown with broken lines. The emission maxima are listed in *Table 3*. (**D**) Normalized fluorescence decay of the hH$_v$1-Acd proteins in Buffer-H2 (gray). The Acd amino acid decays in selected solvents are also shown for reference. Q56Acd and C107Acd decays are shown in red and black, respectively.

The online version of this article includes the following source data for figure 4:

**Source data 1.** Original data for the emission fluorescence spectra of acridon-2-ylalanine (Acd) in different solvents showed in *Figure 4*.

**Source data 2.** Original data for the lifetime of acridon-2-ylalanine (Acd) in different solvents showed in *Figure 4*.

**Source data 3.** Original data for the emission fluorescence spectra of human voltage-gated proton channel (hH$_v$1)-acridon-2-ylalanine (Acd) showed in *Figure 4*.

**Source data 4.** Original data for the lifetime of human voltage-gated proton channel (hH$_v$1)-acridon-2-ylalanine (Acd) showed in *Figure 4*.

Trp/Tyr residues in the N-terminal domain. Conversely, F247Acd showed the lowest *Ratio A*, consistent with its location in the CC, far from most Trp/Tyr residues (*Figure 5B*).

We next compared *Ratio A − Ratio A$_0$* values with FRET efficiencies predicted by the AlphaFold structural model. We modeled rotamer ensembles for Trp, Tyr, and our Acd sites in the hH$_v$1 structural model using chiLife (*Tessmer and Stoll, 2023*) and calculated donor-acceptor distance distributions. These distributions were then used to calculate FRET efficiencies (details in Appendix 1).

**Table 2.** Emission fluorescence spectrum maximum of acridon-2-ylalanine (Acd) in different solvents.
The wavelength corresponds to the location of the maximum intensity of the emission fluorescence spectrum of the free amino acid dissolved in the corresponding solvent. EtAc: ethyl acetate, OctOH: 1-octanol, ButOH: 1-butanol, EtOH: ethanol.

| Solvent | Emission maximum (nm) |
|---|---|
| EtAc | 402 |
| OctOH | 414 |
| ButOH | 415 |
| EtOH | 414 |
| Water | 421 |
| Buffer-H2 | 420 |

The experimentally determined *Ratio A – Ratio A$_0$* values showed a moderate correlation (Pearson's *r*=0.48) with the calculated efficiencies (*Figure 5F*). Using the predicted intrasubunit FRET efficiencies reduced the Pearson's r value to 0.18 (*Figure 5—figure supplement 3*), suggesting that the measured FRET contains information about subunit organization.

There were two spatially clustered deviations between experiment and model. Positions at the extracellular ends of the VSD helices (Y134 in S2, K125 in S1, and F195 in S4) showed higher experimental FRET than predicted, indicating shorter effective donor-acceptor distances relative to the model. Conversely, positions on the intracellular side of the VSD (K169 in S3 and I217 in S4) showed lower experimental FRET than predicted, implying longer effective donor-acceptor distances. Together, the spatial clustering of residues with similar deviations indicates specific deficits in the AlphaFold model or the failure to quantify the protein dynamics.

## Zinc binding changed the conformation of hH$_v$1

We next asked whether FRET from Trp/Tyr to Acd could report conformational changes upon addition of Zn$^{2+}$, the classical inhibitor of H$_v$1 proton currents in cells (*Cherny and DeCoursey, 1999*; *Musset et al., 2010*; *Jardin et al., 2020*). The two experimental structures of H$_v$1 were resolved with this cation bound to the extracellular side of the VSD (*Takeshita et al., 2014*; *Bayrhuber et al., 2019*). We observed site-specific spectral changes in the hH$_v$1-Acd proteins in the presence of Zn$^{2+}$ (*Figure 6*, *Figure 6—figure supplement 1*). The largest changes occurred for hH$_v$1-K169Acd, where

**Table 3.** The emission fluorescence spectrum maximum of the human voltage-gated proton channel (hH$_v$1)-acridon-2-ylalanine (Acd) proteins.
The wavelength corresponds to the location of the maximum intensity of the emission fluorescence spectrum of the samples.

| hH$_v$1 sample | Emission maximum (nm) |
|---|---|
| A18Acd | 418 |
| Q56Acd | 420 |
| M91Acd | 419 |
| V103Acd | 417 |
| C107Acd | 415 |
| K125Acd | 418 |
| Y134Acd | 416 |
| F159Acd | 418 |
| K169Acd | 415 |
| F195 Acd | 416 |
| I217Acd | 415 |
| F247Acd | 418 |

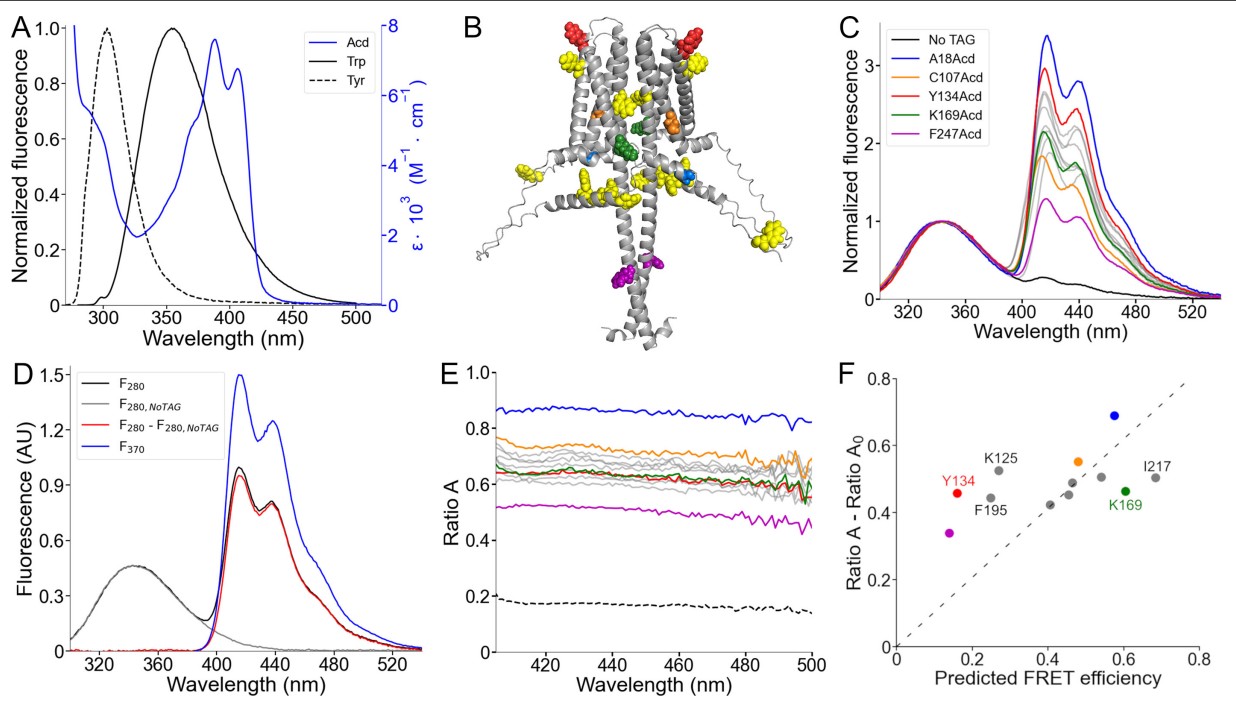

**Figure 5.** Spectral Förster resonance energy transfer (FRET) analysis using Trp and Tyr donors and acridon-2-ylalanine (Acd) as acceptor. (**A**) Fluorescence emission spectra of the intrinsic fluorescent amino acids Trp (black) and Tyr (black, broken lines), along with the Acd absorption spectrum in Buffer-H2 (blue). (**B**) AlphaFold dimer hH$_v$1 structural model with the amino acids colored according to (**C**) and the native tryptophan and tyrosine residues shown as yellow spheres. (**C**) Normalized fluorescence spectra of the purified hH$_v$1-Acd proteins (Excitation = 280 nm). The spectra were normalized by the Trp/Tyr fluorescence. The colors correspond to the indicated hH$_v$1-Acd protein, and the remaining spectra are shown as gray traces in the figure. The No TAG sample (black) contains purified hH$_v$1 protein without any amber stop codon expressed in the presence of Acd and the aminoacyl-tRNA synthetase/tRNA pair. (**D**) Example of the spectral FRET analysis procedure. The fluorescence spectrum of hH$_v$1-Y134Acd obtained when exciting at 280 nm (black, $F_{280}$) minus the normalized spectrum at the same excitation wavelength of the No TAG sample (gray, $F_{280,NoTAG}$) produced the Acd emission spectrum shown in red ($F_{280} - F_{280,NoTAG}$). *Ratio A* values were calculated by dividing this spectrum by the hH$_v$1-Y134Acd fluorescence spectrum obtained by direct excitation at 370 nm ($F_{370}$). (**E**) *Ratio A* traces of the hH$_v$1-Acd proteins colored according to (**C**). The ratio between the spectra of free Acd amino acid excited at 280 and 370 nm (*Ratio A$_0$*) is shown as broken lines. (**F**) Mean *Ratio A – Ratio A$_0$* values (410–480 nm; N=4) as a function of the predicted FRET efficiencies from the AlphaFold hH$_v$1 structural model, colored according to (**C**). The broken line is the best fit to the equation (*Ratio A – Ratio A$_0$*)=m(Predicted FRET efficiency).

The online version of this article includes the following source data and figure supplement(s) for figure 5:

**Source data 1.** Original data for the emission fluorescence spectra of Trp and Tyr and absorption spectrum of acridon-2-ylalanine (Acd) showed in *Figure 5*.

**Source data 2.** Original data for the emission fluorescence spectra of human voltage-gated proton channel (hH$_v$1)-acridon-2-ylalanine (Acd) showed and analyzed in *Figure 5*, *Figure 5—figure supplement 1*, *Figure 5—figure supplement 2*, and *Figure 5—figure supplement 3*.

**Figure supplement 1.** Fluorescence emission spectra of human voltage-gated proton channel (hH$_v$1)-acridon-2-ylalanine (Acd).

**Figure supplement 2.** Fluorescence emission spectra of the acridon-2-ylalanine (Acd) amino acid in solution.

**Figure supplement 3.** Measured FRET efficiency as a function of the predicted intrasubunit FRET efficiency.

1 mM Zn$^{2+}$ caused a decrease in the Trp/Tyr emission and an increase in the Acd emission (*Figure 6A*). This decrease in donor fluorescence, paired with increased acceptor fluorescence, is the hallmark of increased FRET and was corroborated by spectral analysis (*Figure 6B*). Additionally, a slight blue shift of the Trp/Tyr fluorescence was observed for all hH$_v$1-Acd proteins in the presence of Zn$^{2+}$ (*Figure 6— figure supplement 1*). These changes were completely reversed by the addition of EDTA, confirming that the changes arise from a reversible conformational change upon Zn$^{2+}$ binding (*Figure 6A and B*). Large changes in effective FRET efficiency were observed with Acd incorporated at Q56 in the N-terminal domain and C107, F159, and K169 in the VSD (*Figure 6C*). Notably, these positions are intracellular, even though H$_v$1 currents are inhibited by extracellular Zn$^{2+}$ (*Cherny and DeCoursey, 1999*; *Qiu et al., 2016*). These results are consistent with previous studies showing that some extracellular H$_v$1

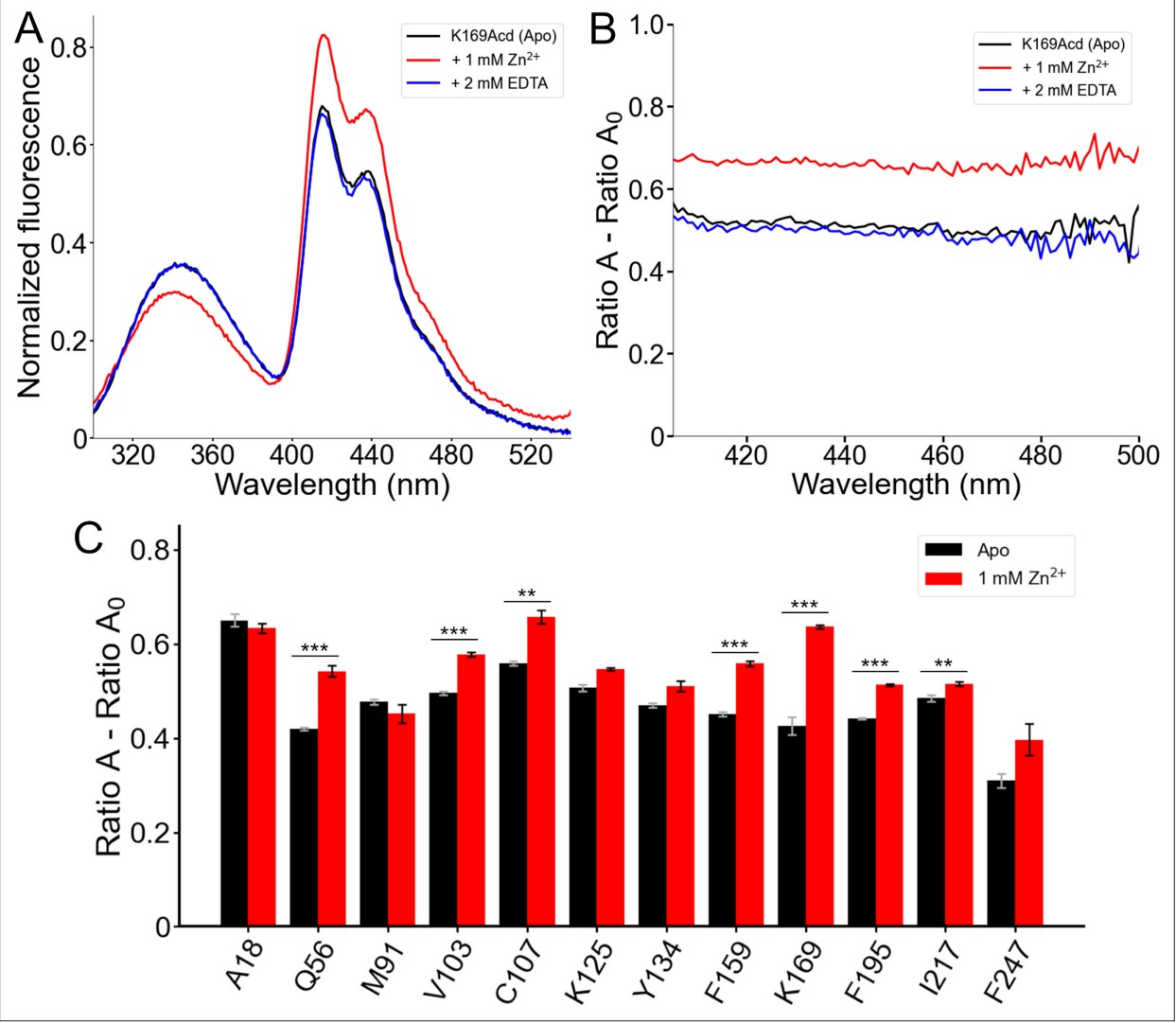

**Figure 6.** Förster resonance energy transfer (FRET) between Trp/Tyr and acridon-2-ylalanine (Acd) reports a conformational rearrangement in response to $Zn^{2+}$. (**A**) Normalized fluorescence emission spectra of human voltage-gated proton channel ($hH_v1$)-K169Acd (Excitation = 280 nm) in Buffer-H2 in the absence of $Zn^{2+}$ (black, Apo), in the presence of 1 mM $Zn^{2+}$ (red), or in the presence of 1 mM $Zn^{2+}$ and 2 mM EDTA (blue). Spectra were normalized by the maximum intensity of the Acd emission spectrum of the same sample excited at 370 nm. (**B**) Spectral FRET analysis of $hH_v1$-K169Acd in Buffer-H2 in the absence of $Zn^{2+}$ (black, Apo), in the presence of 1 mM $Zn^{2+}$ (red), or in the presence of 1 mM $Zn^{2+}$ and 2 mM EDTA (blue). (**C**) Spectral FRET analysis of the $hH_v1$-Acd proteins in Buffer-H2 in the absence (black, Apo) or presence of 1 mM $Zn^{2+}$ (red). N=4, ***$p<0.005$, **$p<0.01$.

The online version of this article includes the following source data and figure supplement(s) for figure 6:

**Source data 1.** Original data for the emission fluorescence spectra of human voltage-gated proton channel ($hH_v1$)-acridon-2-ylalanine (Acd) showed and analyzed in *Figure 6*, *Figure 6—figure supplement 1*.

**Figure supplement 1.** Changes in the fluorescence emission spectra of human voltage-gated proton channel ($hH_v1$)-acridon-2-ylalanine (Acd) in the presence of zinc.

inhibitors, such as $Zn^{2+}$ or AGAP, induce long-range structural changes that propagate to the channel's intracellular vestibule (*Tang et al., 2020*; *De La Rosa et al., 2018*) and persist in detergent micelles (*Bayrhuber et al., 2019*).

## Discussion

Incorporating Acd into $hH_v1$ by GCE opens new possibilities for studying the structure and function of this channel. We systematically incorporated Acd at multiple positions in $hH_v1$ and confirmed protein stability and function after purification. 12 of 14 $hH_v1$-Acd proteins produced acceptable yields upon

purification and were functional, despite being a human membrane protein expressed in *E. coli*. Two factors likely underlie the robustness of our system: (i) an optimized purification protocol for hH$_v$1 (*Carmona et al., 2025a*; *Carmona et al., 2025b*) that offsets the expected reduction in yield associated with GCE (*Speight et al., 2013*; *Liu and Schultz, 2010*), and (ii) the high efficiency and specificity of the evolved aminoacyl-tRNA synthetase for incorporating Acd (*Gordon et al., 2018*; *Zagotta et al., 2021*).

Fluorescence spectroscopy has been used to detect conformational changes in H$_v$1. Sites at the extracellular portion of S4 (*Gonzalez et al., 2010*) and S1 (*Mony et al., 2015*) in the VSD of *Ciona intestinalis* H$_v$1 have been labeled with the fluorophores Alexa488 and tetramethylrhodamine (TAMRA) through cysteine-selective reagents to capture conformational changes during activation using voltage-clamp fluorometry. Similarly, the fluorescent noncanonical amino acid 3-(6-acetylnaphthalen-2-ylamino)–2-aminopropanoic acid (Anap) has been incorporated at several sites in the S4 in hH$_v$1 (*Suárez-Delgado et al., 2023*) to study conformational changes in response to voltage and pH. In these cases, the fluorophore's environmental sensitivity has been used as a readout of conformational changes. By contrast, FRET provides richer structural information due to its distance dependence. Single-molecule FRET has been measured in purified hH$_v$1 labeled with Cy3 and Cy5 to track the VSD movements in response to voltage, pH, cholesterol, and zinc (*Han et al., 2022b*; *Han et al., 2022a*). The small size and minimal linker of Acd, along with its low environmental sensitivity and high photostability, are advantageous for measuring FRET in hH$_v$1 compared with previously used fluorophores.

The high number of FRET donors, four tryptophans and four tyrosines per subunit, limits the utility of the Trp/Tyr-Acd FRET pair as a strategy to measure conformational rearrangements in hH$_v$1. Our library of a dozen single Acd-incorporating constructs, with sites distributed across the entire primary sequence of the protein, opens the door to using Acd as a FRET donor. We have previously used time-resolved transition metal ion FRET (tmFRET), which utilizes a transition metal ion as a FRET acceptor (*Zagotta et al., 2021*; *Gordon et al., 2024*), to measure conformational energetics in the bacterial cyclic nucleotide-gated ion channel SthK (*Eggan et al., 2025*). However, in that work, we incorporated Acd at a single position in the intracellular C-terminal domain of SthK. With Acd incorporated at multiple sites in full-length hH$_v$1, it will be possible to interrogate conformational changes across the protein's different structural domains using Acd as a tmFRET donor to understand its molecular mechanisms.

## Materials and methods
### Protein expression and purification
hH$_v$1 was expressed and purified according to the published protocol with minor modifications (*Carmona et al., 2025a*; *Carmona et al., 2025b*). Briefly, fresh competent BL21-Gold(DE3) cells (Agilent) were co-transformed with the His-EK-hH$_v$1-C107A-C249A.pET15-b (hH$_v$1-Cysless, referred to as No TAG) and MjA9Acd-RS.pDule2 (*Speight et al., 2013*; *Sungwienwong et al., 2017*) plasmids (sequences in *Supplementary file 1*) using the heat shock method. For expressing hH$_v$1-Acd proteins, the codon corresponding to the selected position was changed to the amber stop codon (TAG) by site-directed mutagenesis. All plasmids were sequenced before use. Pre-cultures were grown overnight in LB medium supplemented with 0.2% glucose, 0.4 mg/mL ampicillin, and 0.1 mg/mL spectinomycin at 37 °C and 250 rpm. The next day, cultures were diluted 1:100 in complete autoinduction medium (*Studier, 2005*) supplemented with 0.4 mg/mL ampicillin and 0.1 mg/mL spectinomycin, and then grown at 37 °C and 250 rpm until an OD$_{600}$ of 1.5 was reached. The protein expression level was proportional to the Acd concentration in the culture medium (*Figure 2—figure supplement 2*). We chose 0.6 mM Acd in the culture medium as optimal for hH$_v$1 expression, as higher concentrations only slightly increased the final biomass (*Figure 2—figure supplement 2B*). Cultures supplemented with 0.6 mM Acd were grown overnight at 20 °C and 250 rpm. Cells were then harvested, resuspended in Buffer-H1 (50 mM Tris, 150 mM NaCl, 1 mM benzamidine, 0.17 mg/mL PMSF, pH 8.0) supplemented with 0.5 mg/mL lysozyme, and incubated for 30 min at 4 °C before being flash-frozen. After thawing, the extracts were supplemented with 5 mM MgCl$_2$, fresh protease inhibitors, and 12.5 μg/mL DNase I, incubated for 1 hr at 4 °C, sonicated, and centrifuged at 100,000 g and 4 °C for 1 hr. The membrane pellet was resuspended in Buffer-H1 and stored at –80 °C until further use. After thawing, the membranes were solubilized with 1.5% Anzergent 3–12 (Anatrace) (Anz3-12)

at room temperature for 1 hr. Insoluble material was removed by centrifuging at 100,000 g and 4 °C for 1 hr, and the solubilized membrane extract was incubated with His-Pur Ni-NTA (Thermo Scientific) resin equilibrated with Buffer-H2 (50 mM Tris, 150 mM NaCl, 12 mM Anz3-12, pH 8.0) for 1 hr at room temperature. Resin was collected in a gravity column and washed with 20 column volumes (CV) of Buffer-H2, followed by 16 CV of Buffer-H2 with 90 mM imidazole. The hH$_v$1 protein was eluted with 20 CV of Buffer-H2 with 0.5 M imidazole, was concentrated with 50 kDa cut-off centrifugal filters, and imidazole was removed by FSEC (Ex/Em = 385/450 nm) in an ENrich SEC 650 (Bio-Rad) column with Buffer-H2. The hH$_v$1-containing fractions were concentrated to 0.5–1.5 mg/mL, depending on the amount of purified protein, and then aliquoted, flash-frozen, and stored at –80 °C until use.

## Reconstitution and fluorescence proton flux assays

The hH$_v$1-Acd samples were reconstituted and assayed using a liposome proton flux assay, as described previously, with minor modifications (*Carmona et al., 2025a*). Briefly, soy polar lipid extract (Avati) in chloroform was dried overnight at room temperature in a rotary evaporator under vacuum and then hydrated in Buffer-K (20 mM HEPES, 150 mM KCl, and 1 mM EDTA, pH 7.0) to a concentration of 10 mg/mL. The vesicles were sonicated in a water bath until the solution became translucent, aliquoted, flash-frozen, and stored at –80 °C until further use. Reconstitutions were performed at a 1:100 (protein:lipid, by mass) ratio by mixing 12.5 µg of purified hH$_v$1-Acd with 1.25 mg of liposomes in a final volume of 330 µL Buffer-K containing 8 mM Anz3-12. The mixture was incubated at room temperature for 1 hr, then diluted with 5 mL of Buffer-K and incubated for 30 min. The mixture was further incubated with three cumulative pulses of 100 mg Bio-Beads SM2 (Bio-Rad), each for 1 hr at room temperature. After the final Bio-Beads pulse, the mixture was incubated overnight at 4 °C. The next day, two successive 100 mg additions of Bio-Beads were performed, with a 1 hr incubation at room temperature following each addition. Bio-Beads were removed in a gravity column, the mixture was diluted with 10 mL of Buffer-K, and then centrifuged at 150,000 g for 2 hr at 4 °C. The proteoliposome pellets were resuspended in 250 µL of Buffer-K, aliquoted, flash-frozen, and stored at –80 °C until further use. On the day of the liposome proton flux assays, samples were thawed at 37 °C and then placed on ice. The sample (40 µL) was diluted in Buffer-Na (20 mM HEPES, 150 mM NaCl, 1 mM EDTA, pH 7.0). 9-Amino-6-chloro-2-methoxyacridine (ACMA; Sigma) was added to a final concentration of 2 µM from a 2 mM stock solution in DMSO, and the mixture was incubated at room temperature for 5 min before the fluorescence measurement began. The baseline was recorded for 3 min ($F_{max}$). Subsequently, 10 nM of valinomycin (Cell Signaling) was added from a 10 µM stock solution in DMSO, followed by 1 µM carbonyl cyanide m-chlorophenyl hydrazone (CCCP; Sigma) from a 1 mM stock solution in DMSO to record $F_{min}$. The final volume was 2 mL. The fluorescence signal ($F(t)$) was normalized according to the equation $(F(t)-F_{min})/(F_{max}-F_{min})$. Fluorescence measurements were performed at 5 s intervals with a 2 s integration time using a Fluorolog-3 spectrometer (Horiba) configured to an excitation wavelength of 410 nm (5 nm bandpass) and an emission wavelength of 490 nm (1 nm bandpass) at room temperature.

## Absorption and fluorescence spectra recordings

The absorption and fluorescence spectra were recorded in the indicated solvents or fresh Buffer-H2 at room temperature. Absorption spectra were recorded in a DU 800 Spectrometer (Beckman Coulter) with a wavelength interval of 0.5 nm and a scan speed of 600 nm/min. Fluorescence emission spectra were measured using a Fluorolog-3 spectrometer (Horiba) with an integration time of 0.1 s, 1 nm increments, and excitation and emission slits of 5 nm. For solubilizing Acd in different solvents, a saturated solution of the amino acid was prepared by dissolving 0.5 mg of Acd powder in the respective solvent. Acd spectra were corrected by subtracting a blank of the solvent. hH$_v$1-Acd proteins were measured at concentrations ranging from 180 to 240 nM in Buffer-H2. Each spectrum was corrected by subtracting a blank with the solvent or Buffer-H2 before adding the protein sample to the cuvette.

## Spectral FRET analysis

We used our previously established spectral FRET analysis to remove contamination caused by direct excitation of Acd by 280 nm light (*Zheng et al., 2003*). This method had the added benefit of eliminating errors arising from the recording system's transfer function, variations in the acceptor's quantum yield, or variations in the total concentration of fluorescent molecules. A Trp/Tyr spectrum

was collected from control protein without any TAG codons (no Acd incorporation). This was used to subtract the Trp/Tyr emission spectra collected at 280 nm for each Acd-incorporating protein. This yielded the extracted Acd spectrum, $F_{280}$ ($F_{280}$ - $F_{280,NoTAG}$ in **Figure 5D**), that had two components: the component due to direct excitation of Acd, $F_{280}^{direct}$, and the component due to FRET, $F_{280}^{FRET}$. $F_{280}$ was normalized by the total Acd emission with 370 nm excitation, $F_{370}$. The resulting ratio, termed *Ratio A*, can be expressed as: $Ratio\,A = \frac{F_{280}}{F_{370}} = \frac{F_{280}^{direct}}{F_{370}} + \frac{F_{280}^{FRET}}{F_{370}}$. The direct excitation component, $\frac{F_{280}^{direct}}{F_{370}}$, termed *Ratio A₀*, was measured using free Acd amino acid in Buffer-H2. We then quantified the relative FRET efficiency as $Ratio\,A - Ratio\,A_0 = \frac{F_{280}^{FRET}}{F_{370}}$. This quantity is directly proportional to FRET efficiency. An average *Ratio A* value was calculated between 410 and 480 nm.

## Time-resolved fluorescence measurements

TCSPC measurements were performed in a FluoTime 300 spectrometer equipped with a PMA Hybrid 40 detector and LDH-P-C-375 laser diode head (PicoQuant) at room temperature. Emitted photons were detected at 446 nm with the emission polarizer at the magic angle. The TCSPC measurements were performed in the indicated solvent or fresh Buffer-H2. The hH$_v$1-Acd samples were measured at a concentration of 1 μM and the Acd amino acid at 500 nM. The count rate was always less than 1% of the excitation repetition rate to avoid pile-up distortions of the fluorescence decay. The instrumental response function (IRF) was measured using a dilute Ludox solution in water.

## Acknowledgements

We thank the Oregon State University GCE4ALL (Center for Genetic Code Expansion for All) for their long-standing collaboration. We also appreciate the excellent technical support provided by Dr. James Petersson and Kyle D Shaffer (University of Pennsylvania) with the Acd synthesis. This work was supported by the National Institutes of Health under award numbers R01EY037223 and R35GM145225 to SEG, and R01EY010329 and R35GM148137 to WNZ. EMC is a Pew Latin American Fellow.

## Additional information

### Funding

| Funder | Grant reference number | Author |
|---|---|---|
| National Institutes of Health | R01EY037223 | Sharona E Gordon |
| National Institutes of Health | R35GM145225 | Sharona E Gordon |
| National Institutes of Health | R01EY010329 | William N Zagotta |
| National Institutes of Health | R35GM148137 | William N Zagotta |
| Pew Charitable Trusts | | Emerson M Carmona |

The funders had no role in study design, data collection and interpretation, or the decision to submit the work for publication.

### Author contributions

Emerson M Carmona, Conceptualization, Formal analysis, Investigation, Methodology, Writing - original draft, Writing – review and editing; William N Zagotta, Sharona E Gordon, Conceptualization, Funding acquisition, Methodology, Writing – review and editing

### Author ORCIDs

Emerson M Carmona ![ORCID] https://orcid.org/0000-0003-1332-0372
William N Zagotta ![ORCID] https://orcid.org/0000-0002-7631-8168
Sharona E Gordon ![ORCID] https://orcid.org/0000-0002-0914-3361

Reviewer #1 (Public review): https://doi.org/10.7554/eLife.110161.3.sa1
Reviewer #2 (Public review): https://doi.org/10.7554/eLife.110161.3.sa2
Author response https://doi.org/10.7554/eLife.110161.3.sa3

## Additional files

### Supplementary files
MDAR checklist

Source data 1. Coordinates of the AlphaFold predicted model of the hHv1 dimer (in PDB format).

Supplementary file 1. DNA plasmid sequences in FASTA format.

### Data availability
All data generated or analyzed during this study are included in the manuscript and supporting files; source data files have been provided for all figures.

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

## Appendix 1

## Calculation of the FRET efficiency from the structural model

We calculated the FRET efficiency using the AlphaFold model of the full-length hH$_v$1 (Dataset S1). For each Acd position, the measured steady-state FRET would correspond to the mean FRET efficiency ($E_{FRET}$), which will be proportional to the *Ratio A – Ratio A$_0$* value obtained from the spectral FRET analysis.

For $n$ number of donors and two acceptors in the hH$_v$1 dimer:

$$E_{FRET} = \sum_{i=1}^{n} P_{Ex,i} E_i \tag{1}$$

where $P_{Ex,i}$ is the probability of excitation and $E_i$ is the FRET efficiency of the $i$th donor. $P_{Ex,i}$ is the number of photons absorbed by the $i$th donor divided by the total number of photons absorbed by the total number $n$ of donors, which is quantified by the donor's extinction coefficient. Supposing the same extinction coefficient for each of the Trp ($\varepsilon_W$=5,501 M$^{-1}$ cm$^{-1}$) and Tyr ($\varepsilon_Y$=1,209 M$^{-1}$ cm$^{-1}$) (Supplementary Reference 1), *Equation 1* can be written as:

$$E_{FRET} = \sum_{i=1}^{n} \left( \frac{\varepsilon_i}{\sum_{i=1}^{n} \varepsilon_i} \right) E_i = \frac{1}{n_W \varepsilon_W + n_Y \varepsilon_Y} \left( \varepsilon_W \sum_{i=1}^{n_W} E_{W,i} + \varepsilon_Y \sum_{i=1}^{n_Y} E_{Y,i} \right) \tag{2}$$

where the sum of the total number of Trp ($n_W$) and Tyr ($n_Y$) is equal to $n$. The efficiency of FRET for the $i$th donor can be expressed as the sum of rates of energy transfer to each of the $j$=2 Acd acceptors, $k_{ij}$, divided by the total rates of photon emission. Therefore, the right terms of *Equation 2* can be expressed as:

$$\sum_{i=1}^{n_W} E_{W,i} = \sum_{i=1}^{n_W} \left( \frac{\sum_{j=1}^{2} k_{ij}}{\sum_{j=1}^{2} k_{ij} + (1/\tau_W)} \right) \text{ and } \sum_{i=1}^{n_Y} E_{Y,i} = \sum_{i=1}^{n_Y} \left( \frac{\sum_{j=1}^{2} k_{ij}}{\sum_{j=1}^{2} k_{ij} + (1/\tau_Y)} \right) \tag{3}$$

where the Trp lifetime ($\tau_W$ = 3.1 ns) and tyrosine lifetime ($\tau_Y$ = 3.6 ns) in the absence of donors are considered constant (*Lakowicz, 2006*). Finally, each energy transfer rate $k_{ij}$ is a function of the distance $r_{ij}$ between the $i$th donor and the $j$th acceptor:

$$k_{ij}(r) = \frac{1}{\tau_D} \left( \frac{R_0}{r_{ij}} \right)^6 \tag{4}$$

where the donor lifetime $\tau_D$ and $R_0$ would be $\tau_W$ and 23.5 Å or $\tau_Y$ and 20.9 Å for Trp or Tyr residues, respectively. For determining the $R_0$ values from the spectra overlap, we used an orientation factor value of 2/3, a refraction index of 1.3346 measured for BufferH2 using a refractometer, and quantum yields of 0.12 and 0.13 for Trp and Tyr, respectively (Supplementary Reference 1). Calculating $E_{FRET}$ using the above equations considers a fixed distance between donors and acceptors. A better approach is to consider the distance distributions $P_{ij}(r)$ produced by the different rotameric states of the FRET pair, which can be modeled easily using chiLife (*Tessmer and Stoll, 2023*). Then, *Equation 4* will be replaced by *Equation 5*:

$$k_{ij}(r) = \int_0^\infty P_{ij}(r) \left[ \frac{1}{\tau_D} \left( \frac{R_0}{r} \right)^6 \right] dr \tag{5}$$

