## [Editor Report · eLife Assessment]

This **important** study presents a **convincing** methodological approach to probe the structural features of the full-length human H_v_1 channel as a purified protein. The method is supported by rigorous biochemical assays and spectral FRET analysis, which will interest biophysicists and physiologists studying H_v_1 and other ion channels and membrane proteins. Overall, the work introduces an interesting labeling strategy and provides a methodology that is of value in investigating hH_v_1 in particular and can be extended to other ion channels. The authors also provide preliminary observations regarding conformational changes induced by zinc.

---

## [Referee Report · Reviewer #1 (Public review)]

[Editors' note: this version has been assessed by the Reviewing Editor without further input from the original reviewers. The authors have addressed the comments raised in the previous round of review, shown below.]

In this study, the noncanonical amino acid acridon-2-ylalanine (Acd) was inserted at various positions within the human Hv1 protein using a genetic code expansion approach. The purified mutants with incorporated fluorophore were shown to be functional using a proton flux assay in proteoliposomes. FRET between native tryptophan and tyrosine residues and Acd were quantified using spectral FRET analysis. Predicted FRET efficiencies calculated from an AlphaFold model of the Hv1 dimer were compared to the corresponding experimental values. Spectral FRET analysis was also used to test whether structural rearrangements caused by Zn2+, a well-known Hv1 inhibitor, could be detected. The experimental data provide a good validation of the approach, but further expansion of the analysis will be necessary to differentiate between intra- and intersubunit structural features.

Interestingly, the observed rearrangements induced by Zn2+ were not limited to the protein region proximal to the extracellular binding site but extended to the intracellular side of the channel. This finding agrees with previous studies showing that some extracellular Hv1 inhibitors, such as Zn2+ or AGAP/W38F, can cause long-range structural changes propagating to the intracellular vestibule of the channel (De La Rosa et al. J. Gen. Physiol. 2018, and Tang et al. Brit J. Pharm 2020). The authors should consider adding these references.

Since one of the main goals of this work was to validate Acd incorporation and the spectral FRET analysis approach to detect conformational changes in hHv1 in preparation for future studies, the authors should consider removing one subunit from their dimer model, recalculating FRET efficiencies for the monomer, and comparing the predicted values to the experimental FRET data. This comparison could support the idea that the reported FRET measurements can inform not only on intrasubunit structural features but also on subunit organization.

---

## [Referee Report · Reviewer #2 (Public review)]

This manuscript by Carmona, Zagotta, and Gordon is generally well-written. It presents a crude and incomplete structural analysis of the voltage-gated proton channel based on measured FRET distances. The primary experimental approach is Förster Resonance Energy Transfer (FRET), using a fluorescent probe attached to a noncanonical amino acid. This strategy is advantageous because the noncanonical amino acid likely occupies less space than conventional labels, allowing more effective incorporation into the channel structure.

Fourteen individual positions within the channel were mutated for site-specific labeling, twelve of which yielded functional protein expression. These twelve labeling sites span discrete regions of the channel, including P1, P2, S0, S1, S2, S3, S4, and the dimer-connecting coiled-coil domain. FRET measurements are achieved using acridon-2-ylalanine (Acd) as the acceptor, with four tryptophan or four tyrosine residues per monomer serving as donors. In addition to estimating distances from FRET efficiency, the authors analyze full FRET spectra and investigate fluorescence lifetimes on the nanosecond timescale.

Despite these strengths, the manuscript does not provide a clear explanation of how channel structure changes during gating. While a discrepancy between AlphaFold structural predictions and the experimental measurements is noted, it remains unclear whether this mismatch arises from limitations of the model or from the experimental approach. No further structural analysis is presented to resolve this issue or to clarify the conformational states of the protein.

The manuscript successfully demonstrates that Acd can be incorporated at specific positions without abolishing channel function, and it is noteworthy that the reconstituted proteins function as voltage-activated proton channels in liposomes. The authors also report reversible zinc inhibition of the channel, suggesting that zinc induces structural changes in certain channel regions that can be reversed by EDTA chelation. However, this observation is not explored in sufficient depth to yield meaningful mechanistic insight.

Overall, while the study introduces an interesting labeling strategy and provides valuable methodological observations, the analysis appears incomplete. Additional structural interpretation and mechanistic insight are needed.

---

## [Author Response]

The following is the authors’ response to the original reviews.

**Public Reviews:**

**Reviewer #1 (Public review):**
Interestingly, the observed rearrangements induced by Zn^2+^ were not limited to the protein region proximal to the extracellular binding site but extended to the intracellular side of the channel. This finding agrees with previous studies showing that some extracellular H_v_1 inhibitors, such as Zn^2+^ or AGAP/W38F, can cause long-range structural changes propagating to the intracellular vestibule of the channel (De La Rosa et al. J. Gen. Physiol. 2018, and Tang et al. Brit J. Pharm 2020). The authors should consider adding these references.

We added the suggested references to the Results section.

Since one of the main goals of this work was to validate Acd incorporation and the spectral FRET analysis approach to detect conformational changes in hHv1 in preparation for future studies, the authors should consider removing one subunit from their dimer model, recalculating FRET efficiencies for the monomer, and comparing the predicted values to the experimental FRET data. This comparison could support the idea that the reported FRET measurements can inform not only on intrasubunit structural features but also on subunit organization.

We calculated the predicted intrasubunit FRET efficiency and presented the results in the new Figure 5-figure supplement 3. Pearson’s coefficient decreased from 0.48 for the dimer to 0.18 for the monomer, suggesting the experimental FRET contains information about subunit organization. This was added to the text.

**Reviewer #2 (Public review):**
(1) Tryptophan and tyrosine exhibit similar quantum yields, but their extinction coefficients differ substantially. Is this difference accounted for in your FRET analysis? Please clarify whether this would result in a stronger weighting of tryptophan compared to tyrosine.

We accounted for differences in the extinction coefficients of Trp and Tyr in our calculations, which are detailed in Appendix 1. The assumptions result in a stronger contribution from Trp than from Tyr.

(2) Is the fluorescence of acridon-2-ylalanine (Acd) pH-dependent? If so, could local pH variations within the channel environment influence the probe's photophysical properties and affect the measurements?

The acridone fluorescence, which is the fluorophore in Acd, is not pH-dependent between pH 2 and 9 (Stephen G.S. and Sturgeon R.J. Analytica Chimica Acta. 1977). This was added to the text.

(3) Several constructs (e.g., K125Tag, Y134Tag, I217Tag, and Q233Tag) display two bands on SDS-PAGE rather than a single band. Could this indicate incomplete translation or premature termination at the introduced tag site? Please clarify.

Yes, the additional bands in the WB are due to the termination of translation for the mentioned protein constructs. We added a note in the legend of Figure 2 regarding this point.

(4) In Figure 5F, the comparison between predicted FRET values and experimentally determined ratio values appears largely uninformative. The discussion on page 9 suggests either an inaccurate structural model or insufficient quantification of protein dynamics. If the underlying cause cannot be distinguished, how do the authors propose to improve the structural model of hHv1 or better describe its conformational dynamics?

We understand the confusion about this point. We are not planning to improve the structural model with FRET between Trp/Tyr and Acd. We modified the text to avoid confusion regarding this point. We plan to use Acd as a transition metal ion FRET (tmFRET) donor to study the conformational dynamics of hH_v_1 in the future (Discussion).

(5) Cu^2+^, Ru^2+^, and Ni^2+^ are presented as suitable FRET acceptors for Acd. Would Zn^2+^ also be expected to function as an acceptor in this context? If so, could structural information be derived from zinc binding independently of Trp/Tyr?

Transition metal ion FRET (tmFRET) uses a fluorophore as the donor and a transition metal ion chelator as the acceptor. For FRET to occur between these donor-acceptor pairs, the fluorescence spectrum of the donor must overlap the absorption spectrum of the metal ion (Zagotta et al., eLife. 2021; Zagotta et al., Biophys J. 2024; Gordon et al., Biophys J. 2024). Zn^2+^ does not absorb visible light, so tmFRET cannot occur for this divalent metal.

(6) The investigated structure is most likely dimeric. Previous studies report that zinc stabilizes interactions between hHv1 monomers more strongly than in the native dimeric state. Could this provide an explanation for the observed zinc-dependent effects? Additionally, do the detergent micelles used in this study predominantly contain monomers or dimers?

Our full-length hH_v_1 in Anz3-12 detergent micelles is predominantly a dimer, as demonstrated in the new panel of Figure 3-figure supplement 2. From our data, we cannot compare the effects of zinc between monomers and dimers.

(7) hHv1 normally inserts into a phospholipid bilayer, as used in the reconstitution experiments. In contrast, detergent micelles may form monolayers rather than bilayers. Could the authors clarify the nature of the micelles used and discuss whether the protein is expected to adopt the same fold in a monolayer environment as in a bilayer?

We used Anzergent 3-12 detergent micelles, which stabilize hH_v_1 in solution. We indicated this in the Results and Materials and Methods sections. We are also intrigued by whether protein folding and conformational dynamics differ between detergent micelles and proteoliposomes, but our data do not provide an answer to this question. We found that the proteoliposomes used for measuring the hH_v_1 function don’t have enough Acd signals to record their spectra, preventing us from performing the same FRET measurements between Trp/Tyr and Acd in liposomes. Still, detergent-solubilized hH_v_1 is functional upon reconstitution, demonstrating that its functional folding is not irreversibly altered in micelles.

**Recommendations for the authors:**

**Reviewer #2 (Recommendations for the authors):**
(1) On page 9, the reference to Figure S11 should be corrected to Figure S10.

We thank the reviewer for catching this mistake. It was corrected in the updated version.

(2) On page 9, multiple prior studies describing zinc binding to hHv1 should be acknowledged, for example:Musset et al. (2010), J. Physiol., 588, 1435-1449;Jardin et al. (2020), Biophys. J., 118, 1221-1233.

References were added to the text.

(3) On page 11, the statement "with Acd incorporated ... we can interrogate its gating mechanism in unprecedented detail" appears overly strong relative to the data presented. Another phrasing might be appropriate.

The sentence was changed. It now reads: “With Acd incorporated at multiple sites in full-length hH_v_1, it will be possible to interrogate conformational changes across the protein’s different structural domains using Acd as a tmFRET donor to understand its molecular mechanisms.”